# Journey towards Renewable Energy for Sustainable Development at the Local Government Level: The Case of Hessequa Municipality in South Africa

**Elaine Fouché [1],\* and Alan Brent [1,2]**

1   Department of Industrial Engineering and the Centre for Renewable and Sustainable Energy Studies,
    Stellenbosch University, Stellenbosch 7602, South Africa
2   Sustainable Energy Systems, School of Engineering and Computer Science,
    Victoria University of Wellington, Wellington 6012, New Zealand; alan.brent@vuw.ac.nz
\*   Correspondence: foucheelaine@gmail.com or foucheelaine@sun.ac.za; Tel.: +27-83-609-0116

**Abstract:** The purpose of the research on which this article reports was to investigate how renewable energy forms part of the strategy of a local government, and to evaluate how participatory processes are utilised in the development and communication of this municipal strategy. The research was conducted with Hessequa Municipality, a local authority situated in the Western Cape Province of South Africa. A new aspect of the research was an attempt to consider renewable energy options as part of the municipal strategy, and not as a standalone project. Action research was undertaken and the resulting qualitative data were analysed using thematic analysis. Cognitive mapping was used to display the data and to analyse the causal relationships between different strategic themes. The causal relationships explicitly show that many opportunities for renewable energy solutions are evident in the form of: biomass-to-energy, low-carbon local economic development, small-scale embedded generation, waste-to-energy, and feed-in tariffs. The barriers for implementation are aging infrastructure and financial and legislative constraints. Participatory processes formed an integral part of the strategy formulation, and a unique visualised strategy was developed for communication with local citizens—a first for a local municipality in South Africa.

**Keywords:** participatory processes; renewable energy; strategy; municipality; cognitive map; action research; qualitative; energy autarky; South Africa

---

## 1. Introduction

The case for renewable energy solutions, such as solar energy, wind energy, hydropower and biomass, is robust in South Africa. The country has the third largest solar resource in the world, with an average of more than 2500 h of sunshine per year and average solar radiation levels ranging between 4.5 and 6.5 kWh/m$^2$ in one day [1]. Wind also shows enormous potential in South Africa, with 6422 MW of electricity procured through the Renewable Energy Independent Power Producer Procurement Programme (REIPPPP) and 3776 MW of electricity generation capacity connected to the national grid by June 2018 [2]. According to the Renewable Energy Status Report [1], the cost of solar photovoltaic (PV) technologies decreased by 83% from the REIPPPP Bid Window 1 to R 0.62/kWh, or USD 0.05/kWh. The onshore wind price decreased by 59% over the same period.

South Africa is a signatory of the Kyoto Protocol and the United Nations Framework Convention on Climate Change and, as such, has an obligation to mitigate climate change. The policy landscape enables the implementation of renewable energy technologies and promotes ecologically sustainable development and the use of natural resources [3]. The 1998 White Paper on Energy Policy [4], the 2003

White Paper on Renewable Energy [5] and the 2011 National Climate Change Response White Paper Policy [6] set out the goals and commitment from government to ensure that renewable energy forms a significant part of the South African energy portfolio. The Western Cape province's Green Economy Strategy Framework [7] further highlights that investing in renewable energy technologies is one way of fulfilling the obligation to decrease carbon emissions in South Africa.

The Integrated Resource Plan (IRP) of South Africa is a living plan updated by the Department of Energy, which models different policy scenarios and energy mixes. The latest draft IRP 2018 [8] sets a target to produce a total of 20,000 MW of renewable power (solar PV, wind and concentrated solar power) by 2030, while achieving socio-economic and environmentally sustainable growth. Although South Africa is showing great success with the REIPPPP, driven at a national level, more mechanisms are needed to achieve the IRP objectives, and therefore 200 MW per annum through to 2030 for generation for own use between 1 MW and 10 MW (embedded generation) has been allocated [8]. Implementing renewable energy solutions at a local governmental level could (potentially) be the start of energy self-sufficiency or energy autarky [9], which could hold many benefits, such as reduced transport distances for energy resources through decentralised systems [9], security of energy supply and protection from future price increases [10], local job creation, and local economic growth [9,11]. The Government of South Africa, in Section 151(3) of the Constitution [3], gives municipalities the autonomy to govern the local affairs of their communities within the parameters of national and provincial legislation. The three white papers on energy, renewable energy and climate change response contain numerous policy directives in favour of sustainable development and ensuring energy security through a diversified supply mix [12]. Therefore, municipalities can justify renewable energy implementation initiatives that will drive sustainable development or diversify their supply mix, because they are aligned to national policy directives.

The local municipality of Hessequa saw this potential benefit of energy autarky when a decision was taken to include renewable energy solutions as part of its long-term vision and strategy during an Energy Summit that was held in June 2015 [13]. However, the implementation of renewable energy solutions on a local governmental level creates complex problems, especially when considering the public participative nature of local governmental decision making. When implementing renewable energy technologies many social, institutional, environmental, technical, and economic factors need to be considered within the social-cultural context [14]. A complex problem is also known as a mess [15] or a wicked problem [16]. Ackoff [15] defines a mess as a system of interrelated problems with multiple stakeholders. Pidd [17] (p. 46) quotes Ackoff in stating that a mess is "a set of circumstances in which there is extreme ambiguity and in which there may well be disagreement". Rittel [16] describes a wicked problem as a problem that is difficult or impossible to solve because of its complex interdependencies; where one aspect of the problem being solved causes many other problems. Pidd [17] proposes soft modelling methods to create a representation of such a complex problem situation, to gain a better understanding, to guide strategic decision making and to effectively manage the mess or wicked problem. Soft modelling methods are of a participatory nature, which means that stakeholders are involved as part of the process. Literature shows many successful applications of these soft modelling methods as well as other participatory processes to better manage the contemporary complex problems of our century [18–24]. However, evidence of the application of these methods at a local governmental level in South Africa is limited.

Before starting the planning process for renewable energy projects, the local context should be understood and it should be determined how renewable energy fits into local government's strategy. Strategy is a participatory process of agreeing on priorities and then implementing those priorities [25]. Action research and other participatory research methods play a crucial role in development [26], and therefore this research process provides a platform for becoming part of this journey towards better understanding and agreement on priorities.

Considering the above, the study on which this article reports aimed to investigate how renewable energy solutions form part of the Hessequa strategy, and to understand what is hampering the

implementation of renewable energy solutions at a local governmental level. A new aspect of the research was an attempt to consider renewable energy options as part of the municipal strategy, and not as a standalone project. The article discusses various opportunities for renewable energy within the Hessequa strategy that could be included in future planning. Hessequa Municipality is ready to take on the planning of renewable energy options in collaboration with the Western Cape government and important stakeholders. The second part of the research focused on the participatory processes utilised in the communication and development of the municipal strategy. A need for improved participatory processes has been established and is a focus for local governments in South Africa. Therefore, these participatory processes were evaluated as part of the research. The action research approach followed resulted in a unique application of cognitive mapping and the development of a visualised long-term strategy—a first for a local municipality in South Africa.

## 2. Materials and Methods

### 2.1. Context of the Case Study: Hessequa Municipality

Hessequa Municipality, situated in the Western Cape Province of South Africa, is one of 226 local municipalities in South Africa. Hessequa Municipality shares executive and legislative authority with the Eden District Municipality, making it a Category B municipality. The borders of Hessequa include the inland towns of Heidelberg, Riversdale, and Albertinia and the coastal resorts of Witsand, Jongensfontein, Stillbaai, and Gouritsmond. An estimated 54,351 people (52,642 in 2011 according to Stats SA [27]) reside within Hessequa, of which 78% are urbanised. The population of 52,642 in 2011 consisted of 15,873 households (3.3 residents per household). Formal housing is available to 94.2% of residents and 4.6% of the population have access to informal housing. As far as ethnicity is concerned, most of the population (68.5%) describe themselves as coloured; white people contributed 23.2%; and 7.4% identified themselves as black African [27]. Three formal sectors form the basis of the local economy, namely: finance, insurance, and business services (19.5%); manufacturing (16.3%); and agriculture, forestry, and fishing (15.2%) [28].

The Hessequa municipal governance is made up of an elected council responsible for decision-making and the municipal administration and staff, who implement the work of the municipality. The 17 elected councillors, namely the executive mayor, speaker, mayoral committee, nine ward councillors and proportional representative councillors, make decisions in terms of legislation, such as bylaws and policies, as well as in terms of executive functions such as operations, projects and issues of finance. Every year, Council approves a municipal budget and an Integrated Development Plan (IDP), which sets out how financial resources will be spent and raised and how development should take place in the area. Council is elected every five years. The municipal administration consists of the municipal manager, who leads the municipal staff, and the five directors responsible for: community services; corporate management; financial services; planning, development and environmental services; and technical services. Municipalities in South Africa are responsible for electricity delivery, water supply, sewage and sanitation, storm water systems, refuse removal, fire-fighting services, municipal health systems, decisions regarding land use, municipal roads, municipal public transport, street trading, abattoirs and fresh food markets, parks and recreational areas, libraries and local tourism [29]. Figure 1 shows the location of Hessequa Municipality in the Western Cape province of South Africa.

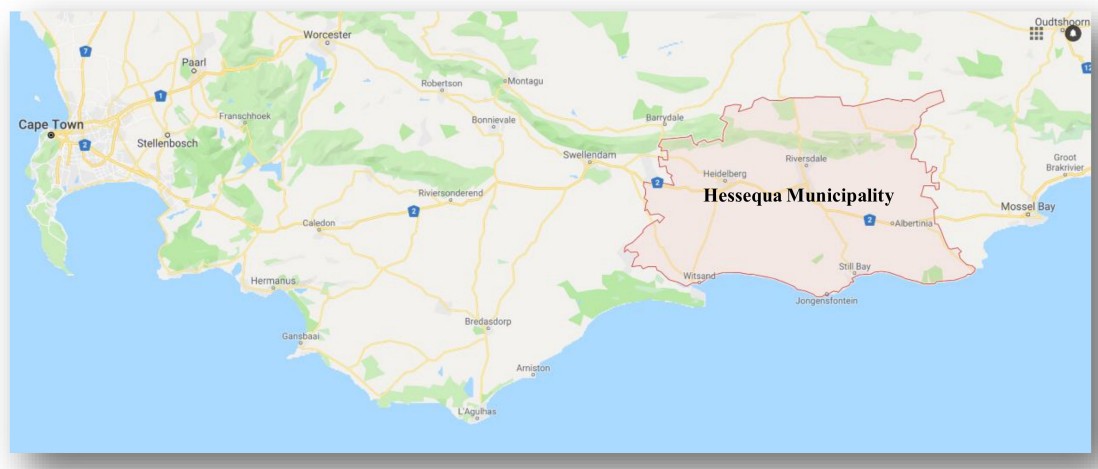

**Figure 1.** Location of Hessequa Municipality, Western Cape, South Africa

## 2.2. Research Methodology

An action research approach [30] was used as a qualitative case study to understand how renewable energy solutions form part of the Hessequa Municipality strategy as well as to evaluate the participatory processes utilised. According to Yin [31] (p. 23), a case study is "an empirical inquiry that investigates a contemporary phenomenon within its real-life context, especially when the boundaries of phenomenon and context are not clearly evident; and in which multiple sources of evidence are used". The qualitative nature of the case study allowed for the main researcher to be part of the case as an observer and a participant to explore and investigate the potential renewable energy options for Hessequa. Action research [32] involves the researcher and the participants as an integral part of the research design to gain understanding of and insight into the worlds of the research participants.

The data and information relevant to the case study were collected through different methods, including the facilitation of workshops with the municipal management and ward committees; participation in several meetings and discussions held with the municipal manager, council, and the technical services manager; observations of IDP and Spatial Development Plan (SDP) meetings held with the nine municipal wards; and analysis of municipal documentation. Primary data, such as transcripts, meeting minutes and field notes, as well as secondary documentation were analysed. Table 1 summarises the interventions that formed part of the research and shows the role of the main researcher in each of these interventions.

**Table 1.** Municipal interventions that formed part of the research.

| Intervention Name and Date | Purpose and Outcome of Intervention | Size of Intervention | Role of Researcher |
|---|---|---|---|
| Hessequa Energy Summit held in Stilbaai on 23 and 24 July 2015 | To establish the possibility of generating and purchasing renewable energy from private producers; the summit concluded a vote of 64 to 36 in favour of pursuing renewable energy opportunities | 250–300 participants | Attendant and observer |
| Community meetings held with communities of Gouritsmond, Stilbaai and Jongensfontein on 11 and 15 September 2015 | To establish unique challenges and development opportunities for Hessequa | 30–50 participants | Researcher did not attend, but received meeting notes from sessions |

**Table 1.** *Cont.*

| Intervention Name and Date | Purpose and Outcome of Intervention | Size of Intervention | Role of Researcher |
|---|---|---|---|
| Meeting with the municipal manager held on 30 March 2016 | To discuss possible future actions on renewable energy projects; a date was set for a workshop with the municipal management team | 4 participants | Attendant; used transcribed data for analysis |
| Hessequa's True North Workshops held on 5 May and 26 May 2016 | As part of the Social Labs driven by the Stellenbosch University School of Public Leadership (SPL), two sessions were facilitated at Hessequa Municipality to determine how the Hessequa Municipality and citizens see their ideal future given the current challenges | 52 participants | Facilitator of the session; used transcribed data and rich pictures for analysis |
| Meeting with the new council after the elections on 30 September 2016 | General discussion to meet the new council and to inform them of the research work done to date | 20 participants | Attendant; used notes from meeting for analysis |
| IDP/SDP meetings held during December 2016 to February 2017 with representatives of the different towns | To discuss the town's specific priorities in terms of strategic direction and spatial development | 10–20 participants | Participant in some of the meetings; also received meeting notes as secondary data |
| Renewable Energy Strategy Meeting on 28 January 2018 | Open discussion to provide feedback on research and to discuss next steps | 9 participants | Presented interim research findings; used meeting notes for analysis |

The Hessequa Energy Summit provided an opportunity for subject matter experts, suppliers of renewable energy solutions, as well as local citizens (consumers) and politicians to come together to share knowledge and ideas regarding the potential of renewable energy solutions within the Hessequa area. An opportunity was also given for participants to work together in groups and to share their ideas and viewpoints. The summit concluded a vote of 64 to 36 in favour of pursuing renewable energy opportunities in Hessequa. Meeting notes and the summit report were used for data analysis.

The Hessequa True North Workshops were facilitated by the main researcher and open questions were asked as to how the participants see their future within Hessequa, and what is currently hampering this ideal future. The people of Hessequa termed this ideal future their 'True North'. Participants worked in groups and were asked to draw a picture of their True North, which was then presented to the other participants. The presentations from the nine groups were voice-recorded and the data were transcribed for analysis. Nine rich pictures were created during this intervention.

The community meetings and IDP/SDP meetings were held with representatives of the different towns within Hessequa to discuss town-specific challenges and development opportunities. During the IDP/SDP meetings, the participants were asked for their opinion regarding renewable energy solutions and where they see town-specific opportunities for renewable energy. Meeting notes and the formal IDP/SDP report were used for data analysis. All the other meetings as indicated in Table 1 were open discussions to determine priorities and next steps. Meetings notes were used for data analysis.

The data collected from the above interventions were used for qualitative content analysis, in combination with cognitive mapping, to understand how renewable energy solutions fit into Hessequa's strategy and to determine which barriers are currently hampering the implementation of renewable energy solutions. A cognitive map is a diagram that consists of concepts (nodes) and arrows that link the different concepts [17]. The main output is a map structured as goals, strategic directions and options (or actions). According to Eden, Ackermann, and Cropper [33], cognitive maps are regarded as a subset of cause maps or causal maps, which are used in the fields of system dynamics [34,35] or systems thinking [36]. System dynamics and systems thinking deal with understanding the behaviour of a human activity system through an understanding of the causal relationships and interdependence of different variables within the system. Literature was then further reviewed to verify barriers as identified during the research process. The second part of the research

was to evaluate how participatory processes form part of the development and communication of the municipality's IDP process.

Municipalities of South Africa are mandated by the Constitution of the Republic of South Africa [3] to focus on growing local economies and providing infrastructure and services through involving citizens in local policy and decision-making. Citizen involvement in local governmental matters is a fundamental democratic right and it occupies a key role in facilitating local democracy and promoting values of good governance. The term used for involving citizens in local governmental matters is 'public participation', which, according to Holmes [37], is a democratic process that provides individuals and groups from the community with an opportunity to influence socio-political and economic conditions for the better. Public participation involves two-way communication, negotiation, and development of mutual understanding, with the ultimate objective of reaching decisions that are supported by the public. During the process of public participation, citizens' concerns, needs and values are incorporated into governmental and corporate decision making [38]. To evaluate these participatory processes, Fouché and Brent's checklist of participatory approaches [39] was used. The checklist summarises the factors necessary for the successful development and implementation of a participatory approach, as found in literature.

## 3. Results

### 3.1. Research Objective 1: Five Main Strategic Themes and Their Causal Links to Renewable Energy Options

Five main strategic themes were determined from analysing the data and developing a cognitive map, as seen in Figure 2. Figure 2 further shows the options for renewable energy highlighted in green and the current barriers to renewable energy options highlighted in orange. The five main strategic themes are to: 1) plan for sustainable infrastructure and innovative service delivery, 2) provide a space for personal and social cohesion, 3) plan for sustainable economic development, 4) plan for environmental conservation, and 5) keep municipal tariffs affordable. Five options for renewable energy solutions became evident during the research period (as marked in Figure 2), namely: a) biomass-to-energy, b) low-carbon local economic development (LED), c) small-scale embedded generation (SSEG), d) waste-to-energy, and e) feed-in tariffs. Each of the strategic themes, linkages to other themes, and opportunities for renewable energy solutions are discussed below.

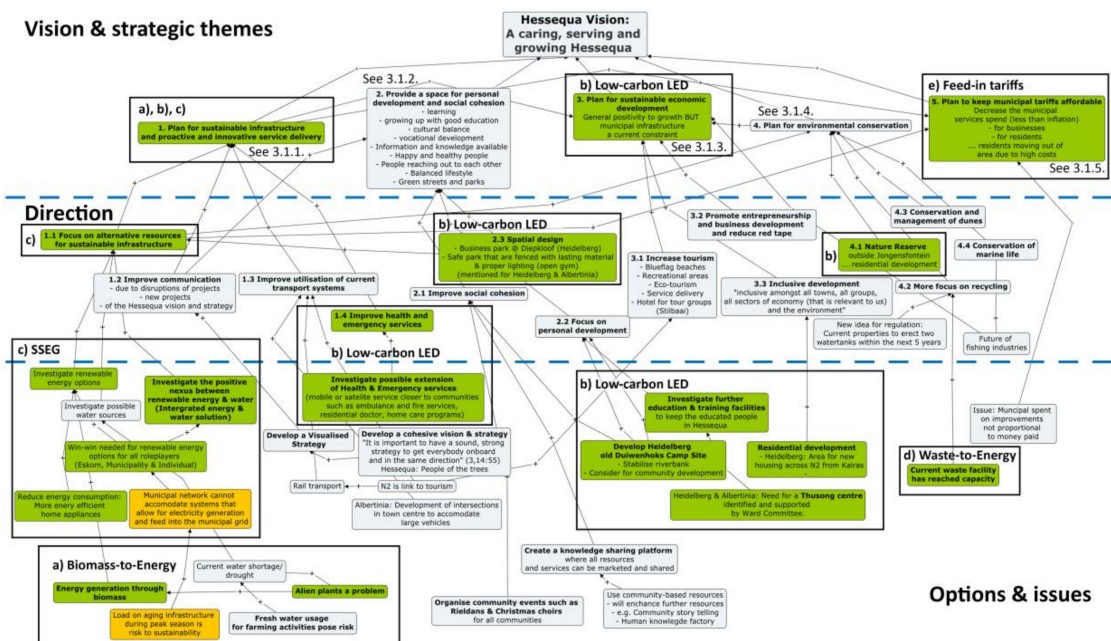

**Figure 2.** Cognitive map of the Hessequa strategic themes with causal links to renewable energy options.

### 3.1.1. Plan for Sustainable Infrastructure and Innovative Service Delivery

The discussions on sustainable infrastructure focused on a need for infrastructure improvement before any new developments can be considered. The sustainable infrastructure mentioned was upgrades to current water reservoirs and dams, sewage systems, the current electricity distribution network and transport systems such as road and rail transport. The load on aging infrastructure, especially during the peak season, posed a problem. In addition, a concern was that the current capacity of the municipal distribution network might not allow for electricity feed into the system, should citizens generate renewable energy from rooftops. What was emphasised was that renewable energy options should be a win-win situation for all role players, namely Eskom (the national utility), the municipality as well as the individuals. The possibility of an integrated energy and water solution emerged, especially in the light of the drought situation facing South Africa [40,41]. No detailed actions were discussed on what this integrated energy and water solutions could entail; however, results from a system dynamics model [42] show that biomass-to-energy in the form of invasive alien plants can contribute towards the nexus of electricity supply and water. Janse van Rensburg's study [42] further showed that solar PV energy is the most attractive renewable energy option in terms of operational and capital cost and definitely shows potential when investigating options for SSEG. Although biomass power is more expensive than solar PV, it shows potential in creating jobs and has a positive environmental impact due to the clearing out of invasive alien plants in the Hessequa area. Biomass power, as a plan for sustainable infrastructure, is interlinked with Theme 4, namely to plan for environmental conservation. The plan for sustainable infrastructure and innovative service delivery (Theme 1) also has a positive link towards sustainable economic development (Theme 3), which is currently constrained by the municipal infrastructure.

A definite link exists between proactive service delivery and the improvement of current health services in Hessequa, especially in the light of an aging population. One possibility is to extend the current health and emergency services to mobile or satellite services closer to the communities, a residential doctor and home care programmes. These possibilities immediately show a link to entrepreneurial and business opportunities (Theme 1 link with Theme 3) and low-carbon LED. The role of renewable energy is evident as an energy source in the applications needed at healthcare facilities, such as lighting, water heating, medical refrigeration and electricity needed for computers, telephones and medical appliances such as microscopes, nebulisers, and centrifuges [43].

Proactive service delivery for the people of Hessequa also means better communication, especially when there are major disruptions due to municipal projects in the area. A link to the need for improved communication was clearly seen where the discussions emphasised that it is "important to have a sound, strong strategy to get everybody on board and in the same direction" [44] (p. 4). Currently, the municipality focuses on its financial plan and IDP as Hessequa's strategy. These documents are lengthy and presented in a way that is not always accessible to some citizens. One specific action from the discussions was to create a visualised strategy, which will provide a visual tool for communicating the Hessequa vision and strategy to the people of Hessequa. The visualised strategy, as a communication strategy, is discussed in Section 3.2.

### 3.1.2. Provide a Space for Personal Development and Social Cohesion

People choose the Hessequa area as a lifestyle destination. This means that they aim to have a safe, balanced, and peaceful stay. It is then understandable that social issues, such as youth unemployment; crime-related activities such as drinking, smoking, and drug abuse; teenage pregnancies; and children not attending school pose a great risk to the residents. Many novel ideas are proposed on how to solve these social problems. However, the people coming up with these ideas do not necessarily understand the cultural background, as one participant rightly said: "We need to adapt our Western thinking to accommodate the needs of the community in a way that is known to them" [44] (p. 5, translated). According to Stats SA, the total unemployment rate in Hessequa is 14.1% and the youth unemployment rate is 18.9% [27]. The highest level of education reached by most of

the population in Hessequa is Grade 12 [28]. Hessequa provides limited opportunities for tertiary education, which means that the person with a Grade 12 certificate will either move out of the area for further studies or will not continue tertiary training due to financial constraints. One definite focus of the municipality is to investigate further educational and training facilities, such as training centres and other knowledge-sharing platforms.

Spatial designs and spaces for social cohesion can be beneficial to personal and social development. Here ideas such as business parks, open parks, recreational areas and areas for community development, which can all contribute towards low-carbon LED, need to be considered. The importance of youth development has been reiterated, which can only be done if communities take ownership and responsibility. Here community leaders can play an important role to build trust and mobilise communities to build a support structure for youths and their parents. Renewable energy plays a crucial role in the sustainability of these spatial designs and spaces as a source of electricity for lighting, computers, and internet connectivity.

### 3.1.3. Plan for Sustainable Economic Development

A portion of the economy of Hessequa is seasonal, with holiday residents owning a large percentage of the properties in Stilbaai. This seasonality can pose a problem for sustainable economic development if not managed. The citizens of Hessequa show a general positivity towards sustainable economic growth in the form of increasing the tourism value chain and the provision of services. Opportunities include tourism infrastructure, transport, signage, basic services, information centres and marketing, developing and improving facilities and human capital [28]. A great focus is on social cohesion, where the people of Hessequa aim to be happy and healthy with a balanced lifestyle. Ideas were given in terms of safe recreational parks (with proper lighting), a business park at Diepkloof in the town of Heidelberg, as well as the development of the old Duiwenshok campsite. The development of basic services, businesses as well as spaces for social cohesion creates further opportunities for renewable energy options (positive link between themes 1, 2, and 3) and low-carbon LED. The Western Cape government is committed to low-carbon LED and is willing to assist in exploring opportunities for business and development related to environmentally friendly and resource-efficient manufacturing [7]. GreenCape is another key stakeholder that has been established to facilitate investment in economic growth and support opportunities related to the renewable energy sector [45]. Most towns in Hessequa are feeling positive towards balanced development, meaning that development can take place, but not at the cost of natural resources or the disturbance of the towns' character. Due to most towns being a retirement destination, the focus of the residents is on the development of spaces for social cohesion, services, and recreational activities. The improvement of health services and the possible extension of health and emergency services are other key objectives for the residents of Hessequa (link between themes 1 and 3).

### 3.1.4. Plan for Environmental Conservation

Environmental conservation has been a theme throughout all the discussions, and the people indicated that they would like to keep the character of the area intact. Some ideas are to develop a nature reserve on an 86-hectare plot outside Jongensfontein; to put regulations in place to enforce more recycling; to collaborate with Cape Nature, the Department of Water Affairs and the Department of Environmental Affairs; better management and conservation of the dunes; and the development of a plan to conserve the marine life. Again, many of these options have a link with Theme 3, namely to plan for sustainable economic development. The link with Theme 1, namely to plan for sustainable infrastructure, has been discussed as part of 3.1.1 above. A threat for Hessequa is the current landfill facility that has reached its capacity. When looking at renewable energy solutions, this current threat can become an opportunity when considering waste-to-energy technology. Waste-to-energy is a viable technology for the disposal of municipal solid waste and energy generation, and has been proven through successfully implemented and operated facilities in Europe and Japan [46]. Drakenstein

Municipality in South Africa successfully motivated a waste-to-energy project with the objectives to minimise waste to landfill, to reduce the carbon footprint on municipalities and to support job creation by unlocking value from waste. This project is currently in a planning phase [47,48].

### 3.1.5. Plan to Keep Municipal Tariffs Affordable

Affordable municipal tariffs is a priority, especially due to the demography of the permanent residents of the area, which are mostly retired and living off a pension fund or citizens earning a low income. In addition, business owners need to keep their businesses running outside of the peak seasons, which is not possible if the municipal tariffs are increasing annually. One concern from residents is that the municipal spend on town improvements is not proportional to the municipal tariffs paid. The municipality aims at keeping the municipal tariff increase below the inflation rate. The municipal tariffs consist of property rates, cost of refuse removal, electricity costs, water costs, and sewage and sanitation costs, of which electricity costs account for 30% of the total municipal bill. Eskom is currently supplying electricity to municipalities, which is then distributed to the properties. The electricity prices have increased by 8% on average annually, which is higher than the current South African Consumer Price Index inflation rate of 4.4% [27]. Water penalties due to the drought is putting further pressure on the municipal bill and it is therefore unrealistic to expect a municipal increase of less than 4.4% under the current circumstances. Renewable energy options in terms of SSEG and feed-in tariffs (see Theme 1) could play an effective role in keeping municipal tariffs affordable. During the time the research was conducted, Hessequa Municipality was in the process of implementing a feed-in tariff policy, which works on the basis of small-scale generators being rewarded for the electricity fed into the municipal grid by discounting their electricity bill.

### 3.2. Research Objective 2: Evaluating the Participatory Nature of Hessequa

To plan for a sustainable energy future, it is argued that approaches inclusive of public participation should be used in the local government sphere. Public participation, if done correctly, can hold many benefits, as indicated by Fouché and Brent [39]: social learning, trust building, knowledge sharing, building a common understanding, changing perceptions and kick-starting ongoing collaboration. Fouché and Brent [39] therefore developed a checklist to ensure the successful development and implementation of participatory approaches. The checklist contains 18 guidelines for consideration when planning a participatory process. These guidelines are referred to in Table 2 as the checklist numbers. The checklist was used primarily to ensure that the participation of the stakeholders was correctly managed and secondly to validate the checklist. The following specific interventions were evaluated during the action research period:

- As part of the social labs driven by Stellenbosch University's School of Public Leadership (SPL), (SPL is committed to community-focused, nationally and internationally competitive academic and professional teaching, research and service delivery in the fields of planning, public management, public policy analysis, development and environmental management for the participative and equitable promotion of sustainable development. SPL is utilising social laboratories in jurisdictions such as Hessequa Municipality, Saldanha Municipality and the Greater Tygerberg Partnership to provide spaces for innovative governance action learning and research (https://www.sun.ac.za/english/faculty/economy/spl/about/history)) two Hessequa True North sessions were facilitated at Hessequa Municipality to determine how they see their ideal future given the current challenges. The first session was held on 5 May 2016 with the municipal management team, consisting of a total of 15 participants. The second session was held on 26 May 2016 with a broader stakeholder group consisting of 37 participants, inclusive of municipal management representatives, ward committee members, municipal council members, representatives from business, and society and representatives from SPL.

- IDP/SDP meetings were held during the period of December 2016 to February 2017 with representatives of the different towns to discuss the towns' specific priorities in terms of strategic direction and spatial development.

Table 2 provides a summary of the findings from the evaluation based on the checklist.

**Table 2.** Summary of evaluation findings on the participative nature of Hessequa.

| What | Checklist Numbers | Evaluation Result |
|---|---|---|
| A holistic and integrated approach was followed. | 1, 12, 13 | Yes |
| A diverse group of stakeholders participated from the start. | 2 | No |
| Careful consideration was given to ways to involve the stakeholders. | 3 | Yes |
| A strong mandate and political support were given for the interventions. | 4 | Yes |
| Good facilitation skills were provided. | 5 | Yes |
| A communication strategy was followed. | 6–9 | Unsure |
| Reflexivity and realism were included as part of the process. | 10 | No |
| The process was underpinned by a philosophy of empowerment, equity, trust and learning. | 11 | Yes |
| Participation was institutionalised. | 14 | Yes |
| Intervention rules were adhered to. | 15–17 | Yes |
| A rationally motivated consensus was reached. | 18 | Yes |

The checklist of participatory approaches [39] highlights in points 1, 12, and 13 that a holistic and integrated approach needs to be followed when developing and implementing a participatory approach. The research focused on following a holistic and integrated approach to ensure that the context and problem situation are firstly understood before starting to evaluate different possible solutions for renewable energy options. The creation of a social lab driven by SPL provided a solid platform to ensure that local and scientific knowledge was integrated. Careful consideration was given to the specific methodologies used and approval from the municipal manager was first established. The methodology used during the Hessequa True North sessions was Checkland's rich pictures [49], which can timeously show the different structures, process aspects and climate of a given situation. Rich pictures are further beneficial because they can isolate key issues quickly, can represent a whole range of stakeholders and can be used to better understand the interconnections of different issues. During the IDP/SDP meetings, cartography or participatory mapping was used as a holistic approach to determine possible future developments for the town. A printout of the current town area and perimeters was used to draw a possible expansion of the town's perimeter and to make notes of the ideas given in the session. In addition, the data from the field notes (with the transcribed data from the workshops) were used to develop the cognitive map, which is another method to visualise and analyse the causal relationships of the different factors.

The participants of the Hessequa True North workshops and the IDP/SDP meetings were not representative of the distribution of the population in terms of ethnicity and age. Although 68% of the population describe themselves as coloured [27], the participants in the room were mostly white men above the age of 40. The voluntary nature of participation makes it difficult for the municipality to involve a more diverse group of stakeholders, yet attention should be given to ways to involve a more diverse group of participants in future discussions. The means used to involve the stakeholders proved successful. Positive feedback on using the rich pictures during the workshops was received, such as that the rich pictures provided a way to progress to the essence of the discussions at a much faster rate. Although nothing new emerged in terms of the challenges the area faces, the rich pictures provided a visual indication of where the urgent matters are. The cartography/participatory mapping used during the IDP/SDP meetings stimulated good discussions in the room on possible future development areas. Both the Hessequa True North workshops and the IDP/SDP meetings had a strong mandate and political support as given in the Constitution and because participation forms part of the

current practices of the municipality. In all cases, the workshops and meetings were facilitated in a professional way.

Points 6 to 9 on the checklist all refer to a communication strategy, which emerged as a major concern throughout most discussions, especially in terms of the development and communication of a shared vision. To ensure buy-in into Hessequa's long-term strategy, a formal communication strategy is needed that will ensure continuity. Here a visualised strategy was created, which will not only frame future discussions in the direction of the strategic themes, but can also invoke creative thinking and ideas. The idea of a visualised strategy emerged from using rich pictures when the future intent of Hessequa was discussed during the True North workshops. The objective of using the visualised strategy (Appendix A: Figure A1) is to create a conducive environment for more strategic discussions during strategic planning sessions at a local government sphere. The visualised strategy can be used as a tool to structure participated discussions between the municipal administration, the municipal council, the public and other important stakeholders, and to provide a powerful and creative way of eliciting what is important to the people of Hessequa.

The feedback received from the workshops and meetings was informal and no evaluation form was used. Throughout the discussions, clear communication was used and a transparent process was followed. Participation within Hessequa Municipality is institutionalised. However, more work is needed to formalise and communicate the rules of such participatory processes. Although the participatory sessions did not explicitly arrive at a rationally motivated consensus, it was clear that the participants felt comfortable with the outcome of the sessions.

## 4. Discussion

### 4.1. How does Renewable Energy Form Part of the Hessequa Strategy, and What Are the Current Constraints for Implementing Renewable Energy Solutions?

Local small renewable energy projects (<1 MW) in South Africa, such as Bethlehem Hydro, eThekwini Landfill Gas, Darling Wind Farm, PetroSA Biogas Power projects, Hessequa Water Purification and George Airport Solar Plant, have proven that successful renewable energy implementation is possible, but to increase the implementation rate the current barriers must be removed and better support is needed from government. Better coordination among policies and institutions is highlighted as a prerequisite for effective renewable energy implementation [50]. Access to modern energy is an important catalyst for economic growth and social equality [51]. The energy systems that drive our economies should be realigned with the ecological systems that define our planetary boundaries to eradicate poverty, create jobs, and sustain growth [52]. The consequences if we fail can be catastrophic, and therefore planning for energy security and sustainable energy should form part of the municipal IDP process. As rightly stated in the Hessequa IDP, local authorities should plan for a longer period than the five years that a council is managing [28]. This planning process should not be a once-off exercise, but should be incremental to handle the many uncertainties and complexities of our time. Renewable energy projects are long-term investments and should be considered as such.

Hessequa Municipality wants to exercise its constitutional right and is ready to take on the challenge to promote sustainable development using renewable energy, but needs national and provincial government support, practical guidance, and funding. Due to the uniqueness of the towns, it was emphasised that the planning of renewable energy projects should be town-specific. The IDP specifically highlights that renewable energy projects should be pursued for the towns of Jongensfontein, Witsand, and Stilbaai, with Stilbaai having the greatest potential for economic growth [28]. The strategic themes determined from analysing the data show a very good parallel with Hessequa's vision of being caring, serving and growing [28]. The discussions did not include details, such as timelines and other specifics, but provided a good idea of what the citizens perceive to be the most important actions and issues in their respective towns. Although renewable energy solutions were mentioned in the discussions, the emphasis was on the socio-economic problems in the area, the citizens' specific needs to maintain and upgrade the current town infrastructure,

their concerns about the high municipal tariffs and possible future increases in tariffs and each town's specific challenges.

The cognitive mapping method provided a way of uniquely structuring the coded data to visually show the interrelatedness between the different themes and data. Although renewable energy solutions were not a key focus for the participants, the causal relationships in the cognitive map show many strengths, weaknesses, opportunities, and threats for renewable energy solutions. Renewable energy solutions that potentially form part of the Hessequa strategy are as follows (marked a) to e) in Figure 2):

(a) Biomass is a potential viable renewable energy source in Hessequa. The possibility of generating renewable energy from the alien plants can offer a solution to the current environmental challenge and has the potential for job creation.

(b) Low-carbon LED is a focus for Hessequa Municipality, as well as the Western Cape government. Future development projects—such as business parks, recreational parks, health and emergency services, and residential developments—create opportunities to incorporate renewable energy technology in the form of solar PV for applications such as lighting, water heating, and electricity generation for small appliances.

(c) SSEG provides a way for home owners and business owners to generate electricity for own use. The preferred technology is solar PV, but technologies such as wind power can also be investigated. To make this a viable option, the constraint of the municipal network and aging infrastructure need to be further investigated in order to accommodate feed into the municipal grid. Hessequa has a linear system, with Eskom supply connecting at the two end points. Feed into the electrical grid will put a constraint on the current system and a more resilient circular distribution system is proposed [13].

(d) Waste-to-energy technology for the reduction and treatment of municipal solid waste can offer a potential solution for the current landfill capacity problem. This option has further advantages in terms of job creation for the Hessequa area.

(e) To keep municipal tariffs at an affordable level, Hessequa Municipality needs to ensure that its citizens are protected against future electricity price increases. The municipality can either generate and sell its own electricity, which is currently constrained by legislation, or offer citizens a viable solution in terms of a feed-in tariff, which can assist in the payback of SSEG investments.

The current drivers and barriers for the implementation of small-scale renewable energy projects in South Africa are further discussed in Section 4.2 below.

### 4.2. Drivers and Barriers to the Implementation of Small-Scale Renewable Energy Projects in South Africa

The cost and affordability of renewable energy remain a challenge [53], due not only to the initial capital investment, but also the high upfront planning and transactional costs [50,53]. However, in South Africa the trajectory of current electricity prices as well as the question of energy security makes it favourable for consumers, businesses, and industries to invest in their own renewable energy generation, such as solar PV. This places municipalities in a dilemma due to the potential revenue loss from electricity sales [12,42]. Kritzinger [54] sees the implementation of feed-in tariffs as a driver for municipalities. Access to renewable energy (or more electricity) does not necessarily reduce the consumer's monthly electricity bill, but could open new opportunities for electricity consumption. The regulatory and legislative environment for renewable energy remains a challenge [1]. Local characteristics of public governance, energy regulation, law enforceability and institutional stability, and policy support mechanisms are some of the key barriers found in literature [53,55]. Eskom has the exclusive right to supply electricity [56] and this restricts access to the national electricity grid [5,57] for local authorities. Municipalities are forced to buy electricity from Eskom and to then sell it to their respective consumers. Also, policies to support renewable projects, especially small-scale projects (<1 MW), are limited and this is hampering the growth in the renewable energy industry [55–60]. Fischer et al. [53] argue that renewable energy requires regulation and incentives to be

financially viable to compete with cheaper technologies. Local authorities may not disregard national or provincial legislation or pass bylaws in conflict therewith [12], yet the regulation of energy needs to be transformed to ensure that local governments do not miss out on opportunities. Macdonald [60] emphasises the importance of moving away from the current trajectory of business as usual in the energy sector in Africa.

Other possible barriers mentioned in literature are the structure and design of the local energy sector [53]; the state and capacity of the current electricity infrastructure [61]; the uncertain political and economic environment of South Africa [57]; the lack of knowledge, skills, and expertise of local governments to embark on a renewable energy journey [22]; and the lack of consumer awareness of the benefits and opportunities of renewable energy [5]. Participatory methods are a definite driver to successfully plan and implement renewable energy solutions, especially in closing the knowledge and skills gap of local governments and their citizens. Participation does not only promote transparency, quality, and comprehensiveness, but also allows for social learning and trust building between stakeholders, among others, as summarised by Fouché and Brent [39]. Krupa and Burch [62] listed effective communication tools, multi-stakeholder participation and incentives for collaboration as mechanisms that may be integral in the journey towards renewable energy. Local authorities need to ensure that their citizens are not only informed, but that they are also actively involved in the planning of a renewable energy future. The evaluation of the participatory processes in Hessequa provided good insights into current practices and generated ideas for improvement, which are discussed in Section 4.3.

### 4.3. How Are Participatory Processes Utilised in the Communication and Development of the Municipal Strategy?

The evaluation of the participatory processes showed that Hessequa has good practices in place to involve its citizens in decision making. Hessequa Municipality has also worked hard to inform its citizens of the benefits of a renewable energy future. One of the shortcomings that was clear from all the interventions is that the Hessequa strategy is not always communicated to all citizens in a way that they can understand. To overcome this shortcoming, the development of a visualised strategy emerged during the action research period from the idea of using rich pictures [49] as a participatory technique. The visualised strategy will be beneficial to frame future discussions of local governmental objectives, and will remain the constant when a new municipal council is elected, as discussed as one of the barriers in Section 4.2. To the knowledge of the researchers, the visualised strategy is a novel concept for municipalities in the Western Cape of South Africa.

### 4.4. Limitations and Future Research

The research was conducted following an action research approach, where the main researcher acted mainly as a participant in the discussions and municipal meetings. The focus for the municipality was on the Hessequa Strategy, its 'True North', and not only on renewable energy solutions. Due to the strategy focus and open questions asked, the depth of the research in terms of the different renewable energy technologies was limited. In addition, the time allowed for the interventions did not permit for detailed questions to be asked regarding specific objectives and timelines and subject matter experts were not always available during these discussions. Future research is therefore encouraged to solely plan for renewable energy solutions with stakeholders and subject matter experts using more formal scientific methods. A formal participatory decision-making model is proposed to evaluate and prioritise the different renewable energy options.

## 5. Conclusions

Electricity is essential in the drive to develop and grow local municipal areas. The research therefore concluded that renewable energy technology for local authorities, such as Hessequa Municipality, is worth investigating. More sessions with diverse stakeholder groups are needed

to develop a plan for implementing renewable energy solutions. The following conclusions were drawn based on the research:

- Energy governance, energy planning, and climate change are global issues and local authorities have a role to play in this regard. Although municipalities only have authority to regulate within their own areas of jurisdiction, they should exert influence over provincial, national and international action in the energy and climate change sphere. Ways in which municipalities can exert influence include leading by example, facilitating and encouraging private sector efforts, policy and other forms of advocacy [12].

- The research elicited many opportunities and synergies for renewable energy projects as part of the strategy of a local municipality, such as the case of Hessequa Municipality. Potential renewable energy solutions became evident through the discussions of the five strategic themes, namely to 1) plan for sustainable infrastructure and innovative service delivery, 2) provide a space for personal and social cohesion, 3) plan for sustainable economic development, 4) plan for environmental conservation, and 5) keep municipal tariffs affordable. The renewable energy opportunities identified were biomass-to-energy, low-carbon LED, SSEG, waste-to-energy, and feed-in tariffs. Biomass-to-energy provides a solution for the environmental challenge of alien plants. Through renewable energy the electricity need, whether for lighting in recreational parks, applications for satellite healthcare clinics, computers and internet connectivity in training centres or for personal electricity use, to reduce the peak electricity demand during high seasons and to keep municipal tariffs affordable can be fulfilled. Renewable energy can further provide opportunities for low-carbon LED and waste-to-energy can provide a viable solution for the current landfill capacity constraint.

- The participatory nature of municipalities provides a conducive environment for future sustainable development. The research showed that including different stakeholders and researchers in the strategic planning of a municipality provides opportunities to elicit sustainable solutions not necessarily considered by the municipality on its own. The creation of a unique visualised strategy can further enhance communication of the municipal strategy. Although the research was limited to one case study, the results can be used to mobilise other local authorities to follow a similar journey. Future research to plan for renewable energy projects for sustainable development and using a participatory decision-making process is recommended.

**Author Contributions:** Elaine Fouché was mainly responsible for conducting the research and writing the article. Alan Brent offered support throughout the study and provided suggestions for and comments on the article.

**Funding:** This research received no external funding.

**Acknowledgments:** The authors acknowledge the funding provided by the Centre for Renewable and Sustainable Energy Studies and the Department of Industrial Engineering of Stellenbosch University, to support the main author and enable the undertaken research and publications. The authors wish to thank all the stakeholders of Hessequa and the SPL who participated in the research and provided valuable inputs to bring the research to conclusion.

**Conflicts of Interest:** The authors declare no conflict of interest.

## Appendix A

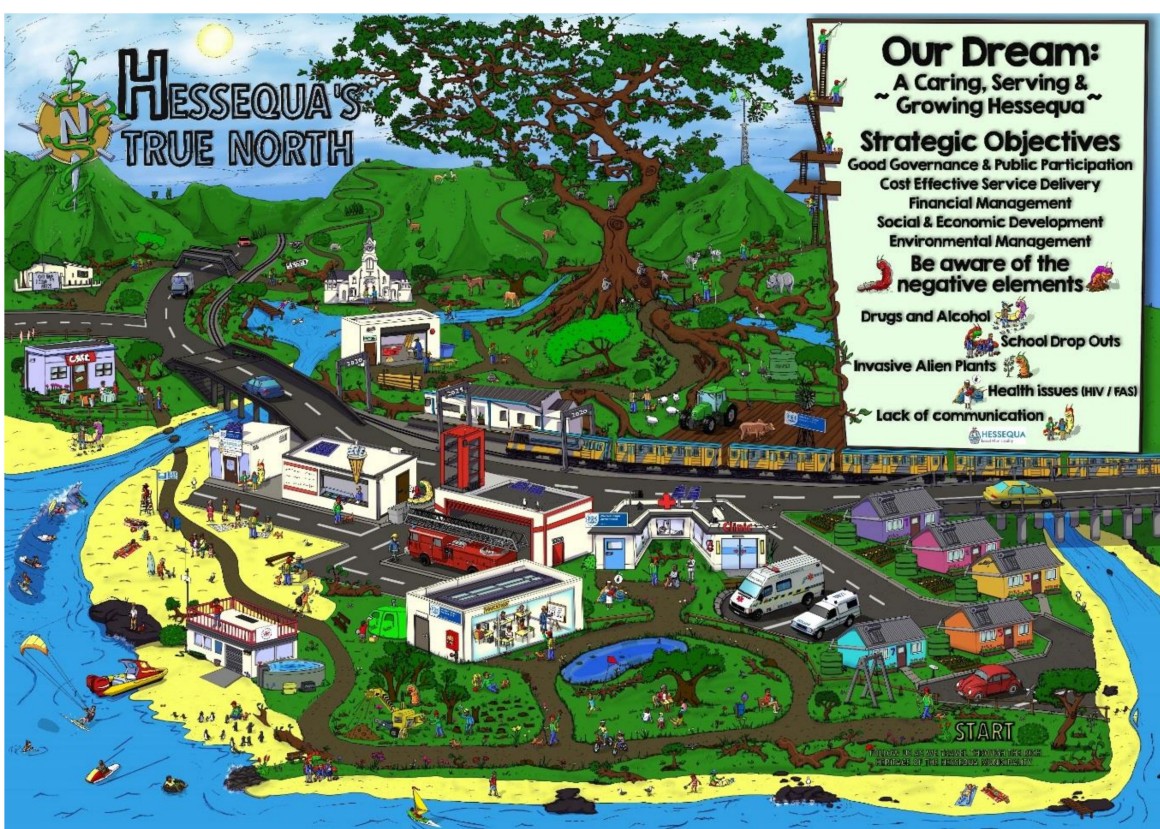

**Figure A1.** Hessequa visualised strategy.

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
