# Peer review of "Journey towards Renewable Energy for Sustainable Development at the Local Government Level: The Case of Hessequa Municipality in South Africa"

_sustainability, doi:10.3390/su11030755_

Round 1

Reviewer 1 Report

The subject, viewpoint and methods of this research are interesting.

It is very meaningful to complete a thesis based on a practical project. However, this method lacks a certain amount of solidity in the overall structure of the paper. The study focuses on the case, but as a result, it should meet the purpose and theme of the paper. Qualitative research methods are hard to define logically as compared to the fields of interest or quantitative research methods.

To overcome these limitations, more clear and systematic approaches are needed.

First, in the introduction, the contents should be supplemented so that the purpose, content and conclusion of the overall study can be read.

Second, The content and conclusion of the overall paper do not meet the research questions. Please redefine the purpose of your paper and research questions.

In the conclusion and discussion of the research, the answers to the research questions are inconsistent and lack of explanation.

Third, for the research method, define more clearly the parts of the participants and supplement the concrete explanation. It is necessary to consistently summarize the size and content of each meeting mentioned in table 1.

(136: systems thinking --> system thinking)

Fourth, add comma

150: 54 351 --> 54,351

152: 52 642 --> 52, 642

152: 15 873 --> 15, 873

Fifth, A description of the connection between 4.2. And 4.3. should be added.

4. 3. why did you evaluate? Add a description of what the checklist numbers mean in table 2.

Sixth,

Figure 2. Hesseque Visualized Strategy & Planning (Change the title)

Storyboard should display the sequence of service or subject to know the sequence.

Author Response

Dear Reviewer,

Thank you for the valuable inputs regarding the manuscript. We have considered all comments and updated the manuscript accordingly. We believe that the improvements made to the updated manuscript will ensure that the article is of high publication quality. The manuscript is better structured and better showcase the purpose of the research which was to investigate how renewable energy forms part of a local government’s strategy and evaluate how participatory processes are utilised in the development and communication of this strategy.

Please note that due to the considerable amount of changes in the manuscript, we turned off Track Changes, but we indicated all the changes in the manuscript with comments. The manuscript was also reviewed by a language editor and we have included the certificate.

Responses to your comments are attached in PDF format.

Kind regards

The Authors

Reviewer 2 Report

It is a good article for publication

Author Response

Dear Reviewer,

Thank you for the good faith you have in the article.  We have considered the comments of the other reviewers and updated the manuscript accordingly. We believe that the updated manuscript is of high publication quality. The manuscript is better structured and better showcase the purpose of the research which was to investigate how renewable energy forms part of a local government’s strategy and evaluate how participatory processes are utilised in the development and communication of this strategy.

Please note that due to the considerable amount of changes in the manuscript, we turned off Track Changes, but we indicated all the changes in the manuscript with comments. The manuscript was also reviewed by a language editor and we have submitted the certificate.

Kind regards

The Authors

Reviewer 3 Report

Dear authors,

I have a mixed opinion about this article. On the one hand, it is very logical, consistent and well-structured. The topic could be of interest to the readers of the journal. On the other hand, it has a number of shortcomings that do not allow to evaluate its scientific and practical value fairly. In this regard, make the following improvements, please:

1. Improve the quality of figure 1.

2. I am not sure that figure 2 should be included in the article. You can just give a link to it.

2. Renumber the subsections in section 4.2. Instead of 1,2,3... should be 4.2.1, 4.2.3, etc.

3. Before subsection 4.2.1. it is recommend to add a scheme that connects strategic themes (the following subsections) and strategic goals (lines 184-189). May be such information is in Figure 1, but i can not read it.

4. Paragraphs 4.2.2-4.2.4. should be described in more details.

5. It is recommended to write the results of the study in terms of bullets in conclusions section.

Author Response

Dear Reviewer,

Thank you for the valuable inputs regarding the manuscript. We have considered all comments and updated the manuscript accordingly. We believe that the improvements made to the updated manuscript will ensure that the article is of high publication quality in terms of scientific and practical value. The manuscript is better structured and better showcase the purpose of the research which was to investigate how renewable energy forms part of a local government’s strategy and evaluate how participatory processes are utilised in the development and communication of this strategy.

Please note that due to the considerable amount of changes in the manuscript, we turned off Track Changes, but we indicated all the changes in the manuscript with comments. The manuscript was also reviewed by a language editor and we have included the certificate.

The responses to your comments are attached in PDF format.

Kind regards

The Authors

Reviewer 4 Report

Language:

I feel that it can be further improved, maybe by a native speaker.

Abstract:

1)      L13-27: The abstract must state the innovation in the conducted research in two or three sentences.

Keywords:

2)      L28-29 – Please find such words which are not in the title, this way search engines of the web will find your manuscript with higher probability.

Introduction:

3)      L58-60: add missing references

4)      The introduction does not indicate innovation in the conducted research. Innovation and novelty should be added in introduction section.

Objective of the paper:

5)      where is located the local municipality of Hessequa? Please add location on Figure.

6)      L80-100: these lines, in my opinion, should not be in this section. This is not objective of paper. Please move to the introduction or other section.

Results:

7)      L145-172: In my opinion, section 4.1. should be moved to the Materials and methods section. This is not result of research.

8)      L145-172: see also my remark no. 5

9)      L490 and L493: Figure 1 and Fig. 2 should be in the text. Now they are in Supplementary Materials.

Discussion:

10)   L408-409: energy autarky is important to different countries. For South Africa also. In this context see https://doi.org/10.15244/pjoes/74129

Conclusions:

11)   In conclusion the authors should answer the research questions from section 2.

12)   What are the weaknesses and limitation of the research carried out?

Author Response

Dear Reviewer,

Thank you for the valuable inputs regarding the manuscript. We have considered all comments and updated the manuscript accordingly. We believe that the improvements made to the updated manuscript will ensure that the article is of high publication quality. The manuscript is better structured and better showcase the purpose of the research which was to investigate how renewable energy forms part of a local government’s strategy and evaluate how participatory processes are utilised in the development and communication of this strategy.

Here are our responses to your comments. Please note that due to the considerable amount of changes in the manuscript, we turned off Track Changes, but we indicated all the changes in the manuscript with comments. The manuscript was also reviewed by a language editor and we will submit the certificate with the updated manuscript.

The responses to your comments are attached in PDF format.

Kind regards

The Authors

Round 2

Reviewer 1 Report

Thank you for your efforts

Reviewer 3 Report

Dear authors,

Considerable efforts have been made to correct the comments of the reviewers. The quality of the article has increased significantly and now it meets the requirements of the Sustainability journal.

Reviewer 4 Report

The article after improvement has gained on quality. The manuscript has gone through a revision as compared to the earlier version.
Currently, the article has a readable form. I recommend it for publication.